# Association between glucose-to-lymphocyte ratio and in-hospital mortality in acute myocardial infarction patients

**Jing Liu** [1]*, **Xiaogang Hu** [2]

1 Department of Emergency (West Branch), The Second Hospital of Shanxi Medical University, Taiyuan, Shanxi, P.R. China, 2 Department of Internal Medicine, Shanxi Agricultural University Hospital, Taiyuan, Shanxi, P.R. China

* liujingEmer@outlook.com

**Data Availability Statement:** The datasets used and/or analyzed during the current study are available from the MIMIC-IV database, https://mimic.physionet.org/iv/.

## Abstract

### Background

Inflammation is involved in the development and progression of atherosclerosis. Recent studies indicated that glucose-to-lymphocyte ratio (GLR) level were significantly associated with the risk of mortality from inflammatory diseases, and showed a specific prognostic value. Herein, this study intended to explore the association between GLR level and in-hospital mortality in patients with acute myocardial infarction (AMI), and evaluate the predictive value of GLR on AMI prognosis.

### Methods

Data of patients with AMI were extracted from the Medical Information Mart for Intensive Care IV (MIMIC-IV) database in 2012–2019 in this retrospective cohort study. Univariate COX proportional hazard model was used to screen covariates. The associations between GLR and in-hospital mortality were evaluated using univariate and multivariate COX proportional hazard models. Subgroup analysis of age, gender, vasopressor use, SOFA scores, renal replacement therapy, coronary artery bypass graft, and β blockers use were performed. The evaluated index was hazard ratios (HRs) and 95% confidence intervals (CIs). In addition, the predictive performance of GLR, glucose, and lymphocytes on in-hospital mortality was assessed respectively.

### Results

Among eligible patients, 248 (13.74%) died in the hospital. After adjusting for covariates, we found that a higher GLR level was associated with an increased risk of in-hospital mortality [HR = 1.70, 95%CI: (1.24–2.34)]. This relationship was also found in patients who were male, aged ≥65 years old, did not have renal replacement therapy, coronary artery bypass graft, or β blockers, used vasopressor or not, and whatever the SOFA scores (all *P*<0.05). Moreover, the predictive performance of GLR on in-hospital mortality seemed superior to that of glucose or lymphocytes.

**Funding:** The author(s) received no specific funding for this work.

**Competing interests:** The authors have declared that no competing interests exist.

## Conclusion

GLR may be a potential predictor for AMI prognosis, which provided some references for identifying and managing high-risk populations early in clinical.

## Introduction

Acute myocardial infarction (AMI) is the myocardial necrosis most often caused by persistent ischemia and hypoxia, resulting from rupture or erosion of atherosclerotic plaques [1]. As a common disease in intensive care units (ICUs), AMI is also a significant cause of acute mortality worldwide [2]. Although the application of interventional therapy as well as the improvement of drugs have partly improved the prognosis of AMI patients, the number of hospitalized AMI patients has increased while the in-hospital mortality has not decreased in the past decade [3, 4]. It is critical to measure biomarker concentrations soon after the onset of AMI, which might contribute to the early identification of patients at high risk of poor results and help further clinical treatment [5]. Regrettably, current indicators for AMI prognosis, such as C-reactive protein and brain natriuretic peptide, are slow and costly [6]. Therefore, uncovering novel predictors that are faster and easier to access is imperative.

Inflammation is involved in the development and progression of atherosclerosis [7]. Lymphocyte count plays a crucial role in regulating inflammatory responses at all stages of atherosclerosis [8] and is associated with AMI occurrence and prognosis [9]. In the early stage of AMI, hyperglycemia promotes prethrombotic status, increases inflammation and sympathetic nervous system activity, worsens endothelial function, induces oxidative stress to release reactive oxygen species, and thus exacerbates coronary artery damage [10, 11]. Paolisso et al. [12] showed that hyperglycemia was associated with higher mortality in patients with AMI, which was independent of diabetes status. The glucose-to-lymphocyte ratio (GLR) is a new marker of inflammation load. Studies have indicated that a higher level of GLR were significantly associated with an increased risk of mortality from inflammatory diseases, such as cancer and acute respiratory distress syndrome (ARDS), and showed a specific prognostic value [13–15]. However, no study has investigated the association between GLR and mortality risk in patients with AMI.

Herein, this study intended to analyze the association between GLR level and the risk of in-hospital mortality in patients with AMI based on the Medical Information Mart for Intensive Care IV (MIMIC-IV) database and evaluated the predictive value of GLR on AMI prognosis to screen out a powerful marker to assist AMI risk stratification.

## Methods

### Study design and population

Data in this retrospective cohort study were extracted from the MIMIC-IV database, which builds upon the success of MIMIC-III and incorporates numerous improvements over MIMIC-III (https://mimic.mit.edu/docs/iv/). MIMIC-IV contains actual hospital stays for patients admitted to a tertiary academic medical center in Boston, MA, USA, between 2012 and 2019 and is intended to support a wide variety of research in healthcare. Comprehensive information for each patient while they were in the hospital, such as laboratory measurements, medications administered, and vital signs were documented and included in MIMIC-IV [16]. This database is approved by the Institutional Review Boards (IRBs) of Massachusetts Institute of Technology and Beth Israel Deaconess Medical Center (BIDMC) [17]. The requirement of

ethical approval for this study was waived by the IRB of The Second Hospital of Shanxi Medical University because the data was publicly available. The IRB of The Second Hospital of Shanxi Medical University waived the need for written informed consent due to the study's retrospective nature.

Adult patients with AMI who were diagnosed by the International Classification of Diseases, 9th revision (ICD-9) code of 41000–41092, or the ICD, 10th revision (ICD-10) code of I210.0-I214.0 or I219.0 (the ICD code of AMI) were included [6]. The exclusion criteria were as follows: (1) age <18 years old, (2) missing information on glucose and lymphocyte count, and (3) hospitalized in the ICU for less than 24 hours.

## Variable collection

The study variables included age, gender, race, insurance, weight, ventilation use, vasopressor use, renal replacement therapy (RRT), percutaneous coronary intervention (PCI), coronary artery bypass graft (CABG), β blockers use, Charlson Comorbidity Index (CCI), congestive heart failure (CHF), atrial fibrillation (AF), cardiogenic shock (CS), diabetes mellitus (DM), hypertension (identified using ICD-9 codes), systolic blood pressure (SBP), diastolic blood pressure (DBP), mean blood pressure (MBP), respiratory rate (RR), heart rate (HR), temperature, $SPO_2$, Simplified Acute Physiology Score II (SAPS-II), Sequential Organ Failure Assessment (SOFA) score, Glasgow Coma Scale (GCS) score, white blood cell (WBC), neutrophil, platelet, hemoglobin (HB), red cell distribution width (RDW), creatinine, international normalized ratio (INR), prothrombin time (PT), partial thromboplastin time (PTT), blood urea nitrogen (BUN), lactate, bicarbonate, sodium (Na), potassium (K), and chloride.

The GLR was calculated by the formula: GLR = glucose (mg/dL) ÷ lymphocytes count (K/uL). In this study, the median GLR was set as the truncation value (106.14), GLR >106.14 was defined as the high GLR level group, and that <106.14 was defined as the low GLR level group. Clinical information was recorded at the admission of the ICU, and only the data from the first ICU hospitalization after the first admission were collected for multiple hospitalizations.

## Outcome and follow-up

The study outcome was in-hospital mortality. In the MIMIC-IV database, in-hospital information of patients was recorded by the hospital department, and that out-of-hospital information was recorded by the Social Security Bureau. Therefore, the death information of the patients is recorded in self-case. The follow-up started at the time of first ICU admission, and ended when patients discharged/died or at December 31, 2020.

## Statistical analysis

Normally distributed measurement data were described by mean ± standard deviation (Mean ± SD) and a t-test was used for the comparison between two groups. Skewed distributed data were described by median and quartiles [M (Q1, Q3)], and the Wilcoxon rank sum test was used for comparison. The frequency with composition ratio [N (%)] was used to describe the distribution of categorical data, and the chi-square test or Fisher's exact test was used for comparison.

Univariate COX proportional hazard model was used for covariates screening. Univariate and multivariate COX proportional hazard models were established to explore the association between GLR and in-hospital mortality in patients with AMI. Multivariate model adjusted for vasopressor, renal replacement therapy, weight, respiratory rate, $SPO_2$, SAPS-II scores, RDW, PTT, BUN, lactate, coronary artery bypass graft, and β blockers. The restricted cubic spline (RCS) curve was used to reflect the relationship between GLR level and the risk of in-hospital

mortality. The Kaplan-Meier (KM) curve was used to assess the association between GLR level and survival probability of AMI patients. The evaluation index was hazard ratios (HRs) and confidence intervals (CIs). $P < 0.05$ indicates a significant difference.

Missing variables were deleted if the proportion of missing value >20%; otherwise, they were interpolated using the random forest interpolation method (S1 Table in S1 File). Sensitivity analysis on the characteristics of patients before and after interpolation of missing data is shown in S2 Table in S1 File. Statistics analyses were completed using SAS 9.4 (SAS Institute., Cary, NC, USA) and R version 4.2.1 (2022-06-23 ucrt).

## Results

### Characteristics of AMI patients

Fig 1 was the flowchart of the participants screening. A total of 2,022 adult patients with AMI who had the information of both glucose and lymphocytes were initially included. Those who hospitalized in the ICU for less than 24 hours (n = 197) were excluded. Finally, 1,805 patients were eligible. Characteristics of AMI patients with different levels of GLR is shown in Table 1. There were 248 (13.74%) patients died in the hospital. The average age of the total population was 68.78 years old, and 1,196 (66.28%) were males. The median glucose, lymphocytes, and GLR between low GLR level group and high GLR level group were respectively 127 mg/dL vs. 171 mg/dL, 2.03 K/uL vs. 0.85 K/uL, and 66.17 vs. 198.41.

### Association between GLR level and in-hospital mortality

Table 2 showed the covariates associated with the in-hospital mortality in patients with AMI. Age, vasopressor use, RRT, CS, weight, RR, HR, SPO$_2$, SAPS-II scores, SOFA score, CCI, neutrophil, RDW, creatinine, INR, PT, PTT, BUN, lactate, bicarbonate, Na, chloride, CABG, and β blockers usage were all significantly associated with in-hospital mortality (all $P < 0.05$).

We further explored the association between GLR and in-hospital mortality (Table 3). After adjusting for covariates (selected by step-to-step regression based on those that have significant difference in the univariate analysis) including vasopressor, RRT, weight, RR, SPO$_2$, SAPS-II scores, RDW, PTT, BUN, lactate, CABG, and β blockers, we found that a higher GLR level was associated with an increased risk of in-hospital mortality [HR = 1.70, 95%CI: (1.24–2.34)], compared with lower level of GLR.

Fig 2 showed the RCS curve of GLR and in-hospital mortality risk, which demonstrated that with the increase of GLR level, HR value increased simultaneously. When GLR <104.6643, the HR was always below 1, while the HR value crossed 1 and raised above 1 when GLR >104.6643. Nonlinear test results indicated that the RCS curve is nonlinear ($P < 0.0001$).

In addition, we plotted the KM curve of GLR levels and the probability of survival in patients with AMI (Fig 3). From the KM curve, the in-hospital survival rate of the high GLR level group was lower than that of the low GLR level group ($P < 0.0001$), representing a significant difference in survival probability between the two groups.

### Relationship between GLR levels and in-hospital mortality in subgroups of age, gender, vasopressor use, SOFA scores, RRT, CABG, and β blockers

We also explored this relationship in different subgroups (Fig 4). The results revealed that a higher GLR level was associated with an increased risk of in-hospital mortality in patients who aged ≥65 years old, were male, not have RRT, CABG, or β blockers use (all $P < 0.05$). Moreover, whatever the SOFA score was and vasopressor used or not, AMI patients who had higher GLR levels all seemed to have higher risk of in-hospital mortality (all $P < 0.05$).

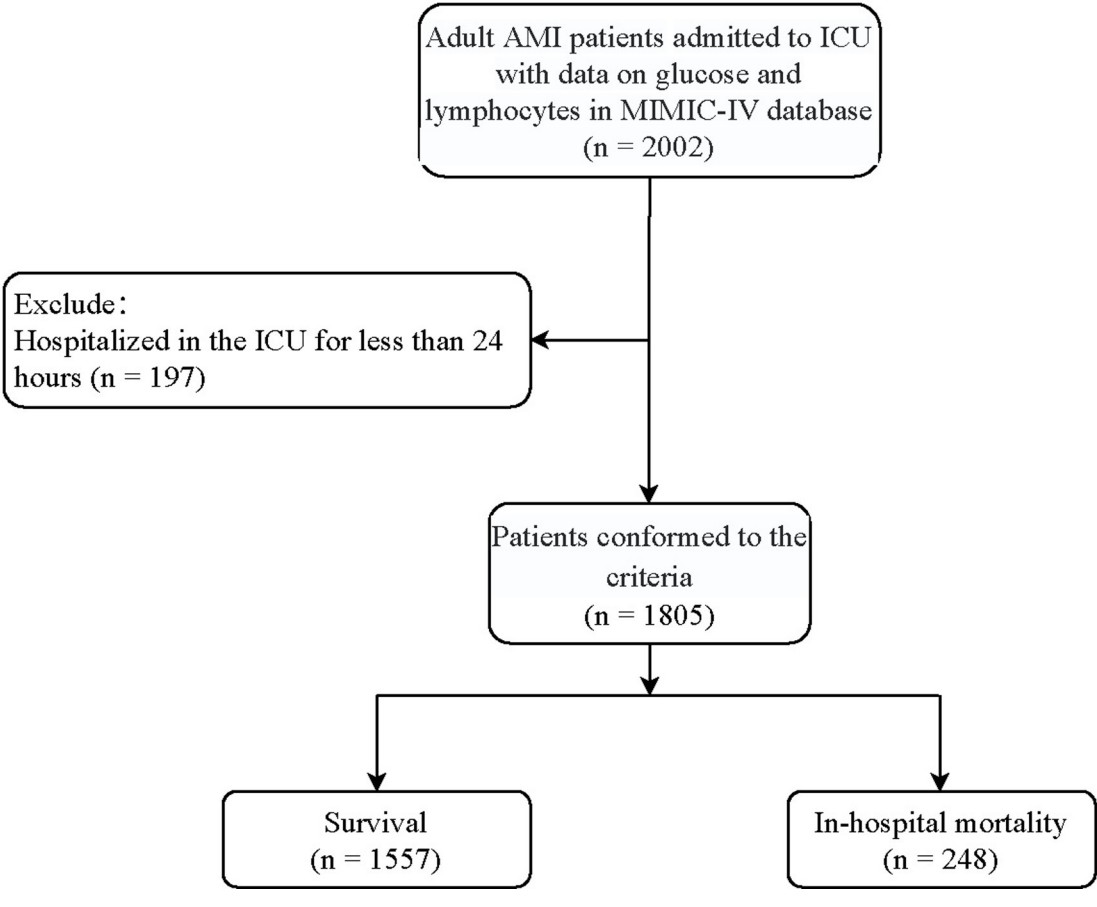

**Fig 1. Flowchart of the participants screening.**

### Predictive performance of GLR, glucose, and lymphocytes on in-hospital mortality

The predictive value of GLR, glucose, and lymphocytes on in-hospital mortality in patients with AMI was further assessed. Fig 5 showed the ROCs of predictive performance of GLR, glucose, and lymphocytes on in-hospital mortality 14 days after the ICU admission. The curve of GLR was consistently higher than that of glucose or lymphocytes, and the graphical trend was basically in equilibrium, indicating the predictive value of GLR may be superior to that of glucose or lymphocytes alone.

## Discussion

In this study, we explored the association between GLR and in-hospital mortality in patients with AMI as well as assessed the predictive value of GLR on AMI prognosis. The results showed that compared with low GLR level, a higher level of GLR was associated with an increased risk of in-hospital mortality. Also, this relationship was found in patients who aged ≥65 years old, were male, not have RRT, CABG or β blockers, had SOFA score <6 or ≥6, and used vasopressor use or not. Moreover, the predictive value of GLR on in-hospital mortality may be superior to that of glucose or lymphocytes.

To our knowledge, no previous study has reported the association between GLR and in-hospital mortality risk in AMI patients. GLR is a recent biomarker that combined the

**Table 1. Characteristics of eligible patients with different levels of GLR.**

| Variables | Total (n = 1805) | Low GLR level (n = 902) | High GLR level (n = 903) | Statistics | P |
|---|---|---|---|---|---|
| Age, years, Mean ± SD | 68.78 ± 12.57 | 67.26 ± 12.46 | 70.30 ± 12.51 | t = -5.17 | <0.001 |
| Gender, n (%) | | | | $\chi^2$ = 0.186 | 0.666 |
| Female | 609 (33.74) | 300 (33.26) | 309 (34.22) | | |
| Male | 1196 (66.26) | 602 (66.74) | 594 (65.78) | | |
| Race, n (%) | | | | $\chi^2$ = 2.947 | 0.400 |
| White | 1165 (64.54) | 565 (62.64) | 600 (66.45) | | |
| Black | 102 (5.65) | 54 (5.99) | 48 (5.32) | | |
| Others | 170 (9.42) | 91 (10.09) | 79 (8.75) | | |
| Unknown | 368 (20.39) | 192 (21.29) | 176 (19.49) | | |
| Insurance, n (%) | | | | $\chi^2$ = 10.123 | 0.006 |
| Medicaid | 77 (4.27) | 42 (4.66) | 35 (3.88) | | |
| Medicare | 940 (52.08) | 436 (48.34) | 504 (55.81) | | |
| Others | 788 (43.66) | 424 (47.01) | 364 (40.31) | | |
| Weight, kg, Mean ± SD | 83.10 ± 18.42 | 84.31 ± 18.40 | 81.89 ± 18.37 | t = 2.80 | 0.005 |
| Ventilation, n (%) | | | | $\chi^2$ = 0.039 | 0.844 |
| No | 118 (6.54) | 60 (6.65) | 58 (6.42) | | |
| Yes | 1687 (93.46) | 842 (93.35) | 845 (93.58) | | |
| Vasopressor, n (%) | | | | $\chi^2$ = 0.577 | 0.448 |
| No | 678 (37.56) | 331 (36.70) | 347 (38.43) | | |
| Yes | 1127 (62.44) | 571 (63.30) | 556 (61.57) | | |
| RRT, n (%) | | | | $\chi^2$ = 37.810 | <0.001 |
| No | 1632 (90.42) | 854 (94.68) | 778 (86.16) | | |
| Yes | 173 (9.58) | 48 (5.32) | 125 (13.84) | | |
| PCI, n (%) | | | | $\chi^2$ = 22.335 | <0.001 |
| No | 1620 (89.75) | 840 (93.13) | 780 (86.38) | | |
| Yes | 185 (10.25) | 62 (6.87) | 123 (13.62) | | |
| CABG, n (%) | | | | $\chi^2$ = 53.304 | <0.001 |
| No | 1566 (86.76) | 730 (80.93) | 836 (92.58) | | |
| Yes | 239 (13.24) | 172 (19.07) | 67 (7.42) | | |
| β blockers use, n (%) | | | | $\chi^2$ = 0.386 | 0.534 |
| No | 1317 (72.96) | 664 (73.61) | 653 (72.31) | | |
| Yes | 488 (27.04) | 238 (26.39) | 250 (27.69) | | |
| CCI, score, M ($Q_1$, $Q_3$) | 3.00 (2.00, 5.00) | 3.00 (2.00, 4.00) | 4.00 (2.00, 6.00) | Z = -9.541 | <0.001 |
| CHF, n (%) | | | | $\chi^2$ = 52.936 | <0.001 |
| No | 924 (51.19) | 539 (59.76) | 385 (42.64) | | |
| Yes | 881 (48.81) | 363 (40.24) | 518 (57.36) | | |
| AF, n (%) | | | | $\chi^2$ = 1.160 | 0.281 |
| No | 1286 (71.25) | 653 (72.39) | 633 (70.10) | | |
| Yes | 519 (28.75) | 249 (27.61) | 270 (29.90) | | |
| CS, n (%) | | | | $\chi^2$ = 52.271 | <0.001 |
| No | 1545 (85.60) | 826 (91.57) | 719 (79.62) | | |
| Yes | 260 (14.40) | 76 (8.43) | 184 (20.38) | | |
| DM, n (%) | | | | $\chi^2$ = 11.161 | <0.001 |
| No | 1079 (59.78) | 574 (63.64) | 505 (55.92) | | |
| Yes | 726 (40.22) | 328 (36.36) | 398 (44.08) | | |
| Hypertension, n (%) | | | | $\chi^2$ = 0.141 | 0.708 |
| No | 648 (35.90) | 320 (35.48) | 328 (36.32) | | |

*(Continued)*

**Table 1.** (Continued)

| Variables | Total (n = 1805) | Low GLR level (n = 902) | High GLR level (n = 903) | Statistics | P |
|---|---|---|---|---|---|
| Yes | 1157 (64.10) | 582 (64.52) | 575 (63.68) | | |
| SBP, mmHg, Mean ± SD | 118.71 ± 21.77 | 116.94 ± 19.73 | 120.49 ± 23.51 | t = -3.48 | <0.001 |
| DBP, mmHg, Mean ± SD | 64.04 ± 15.94 | 62.67 ± 14.55 | 65.41 ± 17.11 | t = -3.66 | <0.001 |
| MBP, mmHg, Mean ± SD | 80.84 ± 16.32 | 80.04 ± 14.77 | 81.63 ± 17.71 | t = -2.08 | 0.038 |
| RR, bpm, Mean ± SD | 18.24 ± 5.21 | 17.06 ± 4.63 | 19.41 ± 5.49 | t = -9.82 | <0.001 |
| HR, bpm, Mean ± SD | 85.67 ± 16.02 | 83.52 ± 14.29 | 87.81 ± 17.33 | t = -5.74 | <0.001 |
| Temperature, °C, Mean ± SD | 36.56 ± 0.64 | 36.50 ± 0.58 | 36.61 ± 0.69 | t = -3.60 | <0.001 |
| $SPO_2$, %, Mean ± SD | 97.55 ± 3.02 | 98.06 ± 2.71 | 97.03 ± 3.23 | t = 7.37 | <0.001 |
| SAPS-II scores, M ($Q_1$, $Q_3$) | 38.00 (30.00, 48.00) | 36.00 (29.00, 44.00) | 41.00 (32.00, 52.00) | Z = -8.323 | <0.001 |
| SOFA score, M ($Q_1$, $Q_3$) | 6.00 (4.00, 9.00) | 5.00 (3.00, 8.00) | 7.00 (4.00, 10.00) | Z = -7.668 | <0.001 |
| GCS score, M ($Q_1$, $Q_3$) | 14.00 (12.00, 15.00) | 14.00 (13.00, 15.00) | 14.00 (10.00, 15.00) | Z = 7.172 | <0.001 |
| WBC, K/uL, M ($Q_1$, $Q_3$) | 12.80 (9.50, 17.10) | 13.30 (10.10, 17.20) | 12.31 (8.60, 17.00) | Z = 3.894 | <0.001 |
| Neutrophil, %, Mean ± SD | 79.26 ± 11.22 | 74.64 ± 10.87 | 83.88 ± 9.54 | t = -19.20 | <0.001 |
| Platelet, K/uL, M ($Q_1$, $Q_3$) | 177.00 (130.00, 236.00) | 169.50 (128.00, 223.00) | 186.00 (131.00, 250.00) | Z = -3.467 | <0.001 |
| HB, g/Dl, Mean ± SD | 10.71 ± 2.50 | 10.68 ± 2.53 | 10.74 ± 2.48 | t = -0.54 | 0.587 |
| RDW, %, Mean ± SD | 14.52 ± 1.98 | 14.26 ± 1.93 | 14.78 ± 2.00 | t = -5.57 | <0.001 |
| Creatinine, mg/dL, M ($Q_1$, $Q_3$) | 1.10 (0.80, 1.70) | 1.00 (0.80, 1.30) | 1.30 (0.90, 2.10) | Z = -11.118 | <0.001 |
| INR, M ($Q_1$, $Q_3$) | 1.37 (1.20, 1.50) | 1.40 (1.20, 1.50) | 1.30 (1.20, 1.50) | Z = 3.173 | 0.002 |
| PT, seconds, M ($Q_1$, $Q_3$) | 14.70 (13.10, 16.90) | 15.10 (13.40, 16.90) | 14.38 (12.90, 16.90) | Z = 3.372 | <0.001 |
| PTT, seconds, M ($Q_1$, $Q_3$) | 33.20 (28.20, 47.10) | 31.70 (27.90, 42.10) | 35.50 (28.60, 53.20) | Z = -5.620 | <0.001 |
| BUN, mg/dL, M ($Q_1$, $Q_3$) | 21.00 (15.00, 34.00) | 17.00 (13.00, 26.00) | 26.00 (17.00, 44.00) | Z = -13.239 | <0.001 |
| Lactate, mmol/L, M ($Q_1$, $Q_3$) | 1.90 (1.41, 2.60) | 1.79 (1.40, 2.40) | 2.00 (1.50, 2.90) | Z = -6.706 | <0.001 |
| Bicarbonate, mEq/L, Mean ± SD | 22.06 ± 4.03 | 22.71 ± 3.39 | 21.42 ± 4.49 | t = 6.92 | <0.001 |
| Na, mEq/L, Mean ± SD | 136.60 ± 4.83 | 136.23 ± 4.11 | 136.97 ± 5.44 | t = -3.27 | 0.001 |
| K, mEq/L, Mean ± SD | 4.42 ± 0.78 | 4.43 ± 0.75 | 4.41 ± 0.81 | t = 0.43 | 0.667 |
| Chloride, mEq/L, Mean ± SD | 103.80 ± 6.23 | 104.51 ± 5.77 | 103.09 ± 6.59 | t = 4.89 | <0.001 |
| ICU hospitalization, days, M ($Q_1$, $Q_3$) | 2.73 (1.54, 4.96) | 2.19 (1.35, 3.76) | 3.19 (1.90, 6.19) | Z = -9.881 | <0.001 |
| Follow-up time, days, M ($Q_1$, $Q_3$) | 6.24 (4.33, 10.81) | 5.74 (4.29, 8.68) | 7.41 (4.63, 12.23) | Z = -6.258 | <0.001 |
| Outcomes, n (%) | | | | $\chi^2$ = 86.285 | <0.001 |
| Survival | 1557 (86.26) | 846 (93.79) | 711 (78.74) | | |
| Mortality | 248 (13.74) | 56 (6.21) | 192 (21.26) | | |
| Glucose, mg/dL, M ($Q_1$, $Q_3$) | 142.00 (117.00, 188.00) | 127.00 (108.00, 152.00) | 171.00 (131.00, 222.00) | Z = -17.182 | <0.001 |
| Lymphocytes, K/uL, M ($Q_1$, $Q_3$) | 1.38 (0.85, 2.06) | 2.03 (1.58, 2.71) | 0.85 (0.58, 1.17) | Z = 31.775 | <0.001 |
| GLR level, M ($Q_1$, $Q_3$) | 106.14 (66.30, 198.41) | 66.17 (48.97, 84.22) | 198.41 (138.84, 312.50) | Z = -36.783 | <0.001 |

GLR: glucose-to-lymphocyte ratio, SD: standard deviation, RRT: renal replacement therapy, PCI: percutaneous coronary intervention, CABG: coronary artery bypass graft, CCI: Charlson Comorbidity Index, M: median, $Q_1$:1st quartile, $Q_3$:3rd quartile, CHF: congestive heart failure, AF: atrial fibrillation, CS: cardiogenic shock, DM: diabetes mellitus, SBP: systolic blood pressure, DBP: diastolic blood pressure, MBP: mean blood pressure, RR: respiratory rate, bpm: beats per minute, HR: heart rate, SAPS-II: Simplified Acute Physiology Score II, SOFA: Sequential Organ Failure Assessment, GCS: Glasgow Coma Scale score, WBC: white blood cell, HB: hemoglobin, RDW: red cell distribution width, INR: international normalized ratio, PT: prothrombin time, PTT: partial thromboplastin time, BUN: blood urea nitrogen, Na: sodium, K: potassium, ICU: intensive care unit.

t: t-test, Z: rank sum test, $\chi 2$: chi-square test, -: Fisher's exact test

inflammatory indicator lymphocyte and blood glucose level to predict prognosis in some diseases. Zhang et al. [14] indicated that GLR is an independent prognostic factor for pancreatic ductal adenocarcinoma patients undergoing curative resection. Zhong et al. [18] also recognized GLR as an independent indicator for the prognosis in patients with pancreatic cancer.

**Table 2. Covariates associated with in-hospital mortality in AMI patients.**

| Variables | HR (95% CI) | P |
|---|:---:|:---:|
| Age | 1.02 (1.01–1.03) | <0.001 |
| Gender | | |
| Female | Ref | |
| Male | 0.92 (0.71–1.19) | 0.511 |
| Race | | |
| Black | Ref | |
| White | 0.87 (0.53–1.42) | 0.581 |
| Others | 0.85 (0.47–1.57) | 0.611 |
| Unknown | 1.15 (0.67–1.97) | 0.603 |
| Insurance | | |
| Medicaid | Ref | |
| Medicare | 1.52 (0.77–2.99) | 0.223 |
| Others | 1.26 (0.64–2.50) | 0.504 |
| Ventilation | | |
| No | Ref | |
| Yes | 2.40 (0.99–5.83) | 0.052 |
| Vasopressor | | |
| No | Ref | |
| Yes | 2.81 (1.98–3.99) | <0.001 |
| RRT | | |
| No | Ref | |
| Yes | 1.54 (1.14–2.09) | 0.005 |
| AF | | |
| No | Ref | |
| Yes | 1.18 (0.90–1.53) | 0.226 |
| CS | | |
| No | Ref | |
| Yes | 1.96 (1.50–2.57) | <0.001 |
| Hypertension | | |
| No | Ref | |
| Yes | 0.82 (0.64–1.05) | 0.120 |
| Weight | 0.99 (0.99–0.99) | 0.045 |
| MBP | 1.00 (0.99–1.01) | 0.859 |
| RR | 1.06 (1.04–1.08) | <0.001 |
| HR | 1.01 (1.01–1.02) | <0.001 |
| Temperature | 0.94 (0.78–1.14) | 0.533 |
| $SPO_2$ | 0.93 (0.90–0.96) | <0.001 |
| SAPS-II scores | 1.04 (1.03–1.05) | <0.001 |
| SOFA score | 1.16 (1.13–1.19) | <0.001 |
| CCI | 1.10 (1.05–1.16) | <0.001 |
| Neutrophil | 1.01 (1.01–1.02) | 0.019 |
| Platelet | 1.00 (1.00–1.00) | 0.280 |
| HB | 0.99 (0.94–1.04) | 0.729 |
| RDW | 1.11 (1.06–1.16) | <0.001 |
| Creatinine | 1.12 (1.05–1.19) | <0.001 |
| INR | 1.09 (1.01–1.18) | 0.040 |
| PT | 1.01 (1.01–1.02) | 0.020 |

(*Continued*)

**Table 2.** (Continued)

| Variables | HR (95% CI) | P |
|---|---|---|
| PTT | 1.01 (1.01–1.01) | <0.001 |
| BUN | 1.01 (1.01–1.02) | <0.001 |
| Lactate | 1.13 (1.09–1.17) | <0.001 |
| Bicarbonate | 0.93 (0.90–0.95) | <0.001 |
| Na | 1.04 (1.01–1.06) | 0.002 |
| K | 0.94 (0.81–1.10) | 0.462 |
| Chloride | 0.98 (0.96–0.99) | 0.013 |
| PCI | | |
| No | Ref | |
| Yes | 0.93 (0.58–1.49) | 0.769 |
| CABG | | |
| No | Ref | |
| Yes | 0.28 (0.15–0.53) | <0.001 |
| β blockers use | | |
| No | Ref | |
| Yes | 0.54 (0.41–0.73) | <0.001 |

AMI: acute myocardial infarction, HR: hazard ratio, CI: confidence interval, Ref: reference, RRT: renal replacement therapy, AF: atrial fibrillation, CS: cardiogenic shock, MBP: mean blood pressure, RR: respiratory rate, HR: heart rate, SAPS-II: Simplified Acute Physiology Score II, SOFA: Sequential Organ Failure Assessment, CCI: Charlson Comorbidity Index, HB: hemoglobin, RDW: red cell distribution width, INR: international normalized ratio, PT: prothrombin time, PTT: partial thromboplastin time, BUN: blood urea nitrogen, Na: sodium, K: potassium, PCI: percutaneous coronary intervention, CABG: coronary artery bypass graft.

Another study demonstrated that GLR before sorafenib treatment was a new prognostic biomarker that may predict survival in advanced hepatocellular carcinoma [15]. Similarly, in the current study, we found that a higher level of GLR was associated with an increased risk of in-hospital mortality in patients with AMI. We further assessed the predictive value of GLR, and compared it with that of glucose or lymphocytes. The results indicated that the predictive performance of GLR may be superior to the other two indexes. Herein, our results may partly provide some references to verify that GLR is a potential good predictor for AMI prognosis.

The exact mechanisms of the association between GLR level and in-hospital mortality risk in patients with AMI is still unclear. Hyperglycemia has been reported to be positively correlated with higher mortality in patients with AMI, which was independent of the diabetes status [12]. In the early stage of AMI, hyperglycemia promotes prethrombotic rate, increases inflammation and sympathetic nervous system activity, worsens endothelial function, induces oxidative stress to release reactive oxygen species, and thus exacerbates coronary artery damage [10,

**Table 3. Association between GLR level and in-hospital mortality in AMI patients.**

| Variables | Low GLR level HR (95% CI) | P | High GLR level HR (95% CI) | P |
|---|---|---|---|---|
| Univariate model | Ref | - | 2.64 (1.95–3.56) | <0.001 |
| Multivariate model | Ref | - | 1.70 (1.24–2.34) | 0.001 |

GLR: glucose-to-lymphocyte ratio, AMI: acute myocardial infarction, HR: hazard ratio, CI: confidence interval, Ref: reference

Multivariate model adjusted for vasopressor, renal replacement therapy, weight, respiratory rate, SPO$_2$, SAPS-II scores, RDW, PTT, BUN, lactate, coronary artery bypass graft and β blockers.

## RCS Curve of GLR and in−hospital mortality risk

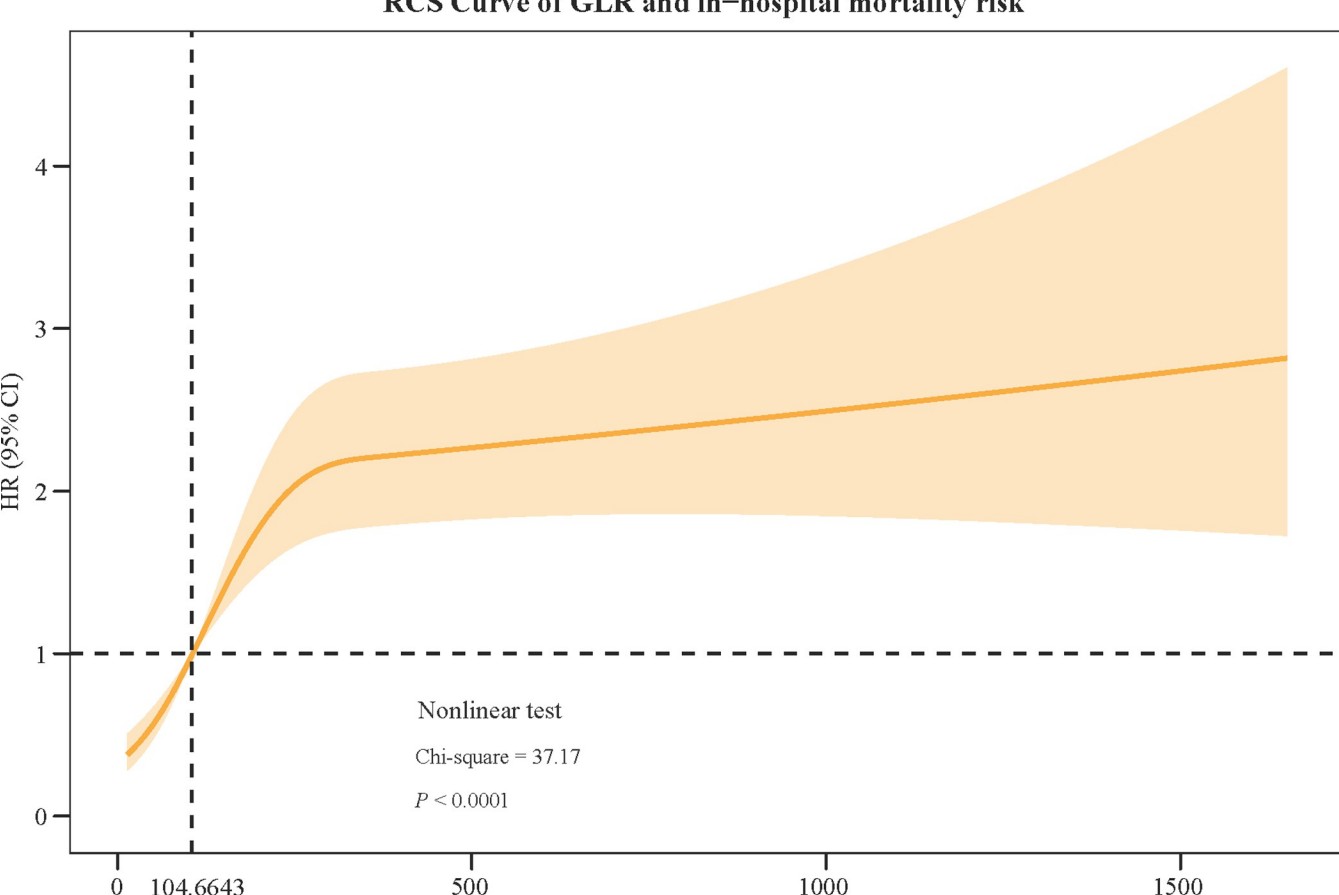

**Fig 2. The RCS curve of GLR and in-hospital mortality risk.**

11]. Accumulating studies have reported that hyperglycemia on admission was positively correlated with poor prognosis during hospitalization [19, 20]. On the other hand, as the indicators of immunity, decreased lymphocyte counts indicate immunity injury [21]. It has been reported that lymphocyte count plays a crucial role in regulating inflammatory responses at all stages of atherosclerosis [8], and is associated with occurrence and prognosis of AMI [9]. Migration of activated lymphocytes to the inflammatory site in vivo, such as the pancreas and lungs, depletes circulating lymphocytes [22]. Therefore, it is of great value to consider that GLR may reflect the depressed immunity and synergistic effect of hyperglycemia in patients with AMI. However, the specific mechanisms of the role of GLR in AMI prognosis are needed further exploration.

According to the subgroup analysis results, higher GLR levels showed a significant association with the increased risk of in-hospital mortality in AMI patients who were male, aged ≥65 years old, not have conventional treatment. In fact, both glucose tolerance and adaptive immune function exhibit significant age-related alterations. A study in old mice showed that T lymphocyte depletion ameliorates age-related metabolic impairments [23]. Similarly, in our study, the higher risk of in-hospital mortality in older AMI patients might be attributed to their poor overall health as well as the increased complications. Therefore, older adults are becoming an increasingly important subpopulation that requires special attention in light of

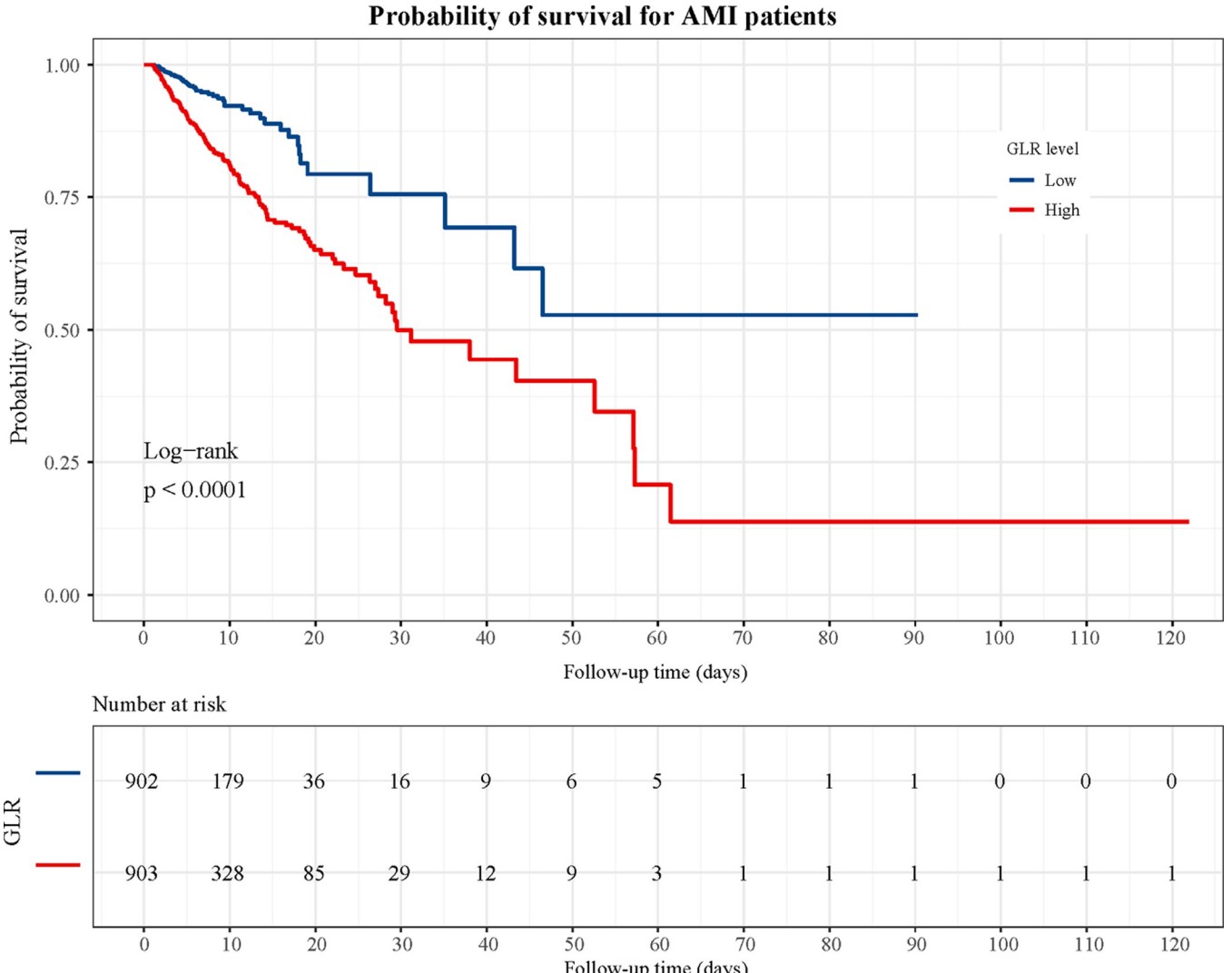

**Fig 3. The KM curve of association between GLR levels and survival probability of patients AMI.**

health and social issues. Besides, AMI is more common in men, and the male gender is also an independent cardiovascular risk factor [24]. However, women and the older patients are significantly underrepresented in clinical trials due to the higher incidence of cardiovascular disease in men and recruitment problems among older persons and women [25, 26]. Females in animal models are also much less readily used because of the potential impact of variation in sex hormone levels during the hormonal cycle [27]. Moreover, we found that among AMI patients without any conventional treatment, higher GLR levels were associated with a higher risk of in-hospital mortality. Marenzi et al. [28] indicated that renal replacement therapy use was associated with a significant increase in in-hospital mortality and a more complicated clinical course in AMI patients. Malmberg et al. [29] also found the in-hospital mortality to be significantly higher when CABG was performed for patients with AMI, compared with patients with stable coronary artery disease. However, another study indicated that β blockers might not be related to the all-cause mortality in patients with AMI [30]. In our study, it seemed that the assessment of GLR is of more significant concern in patients have not received the

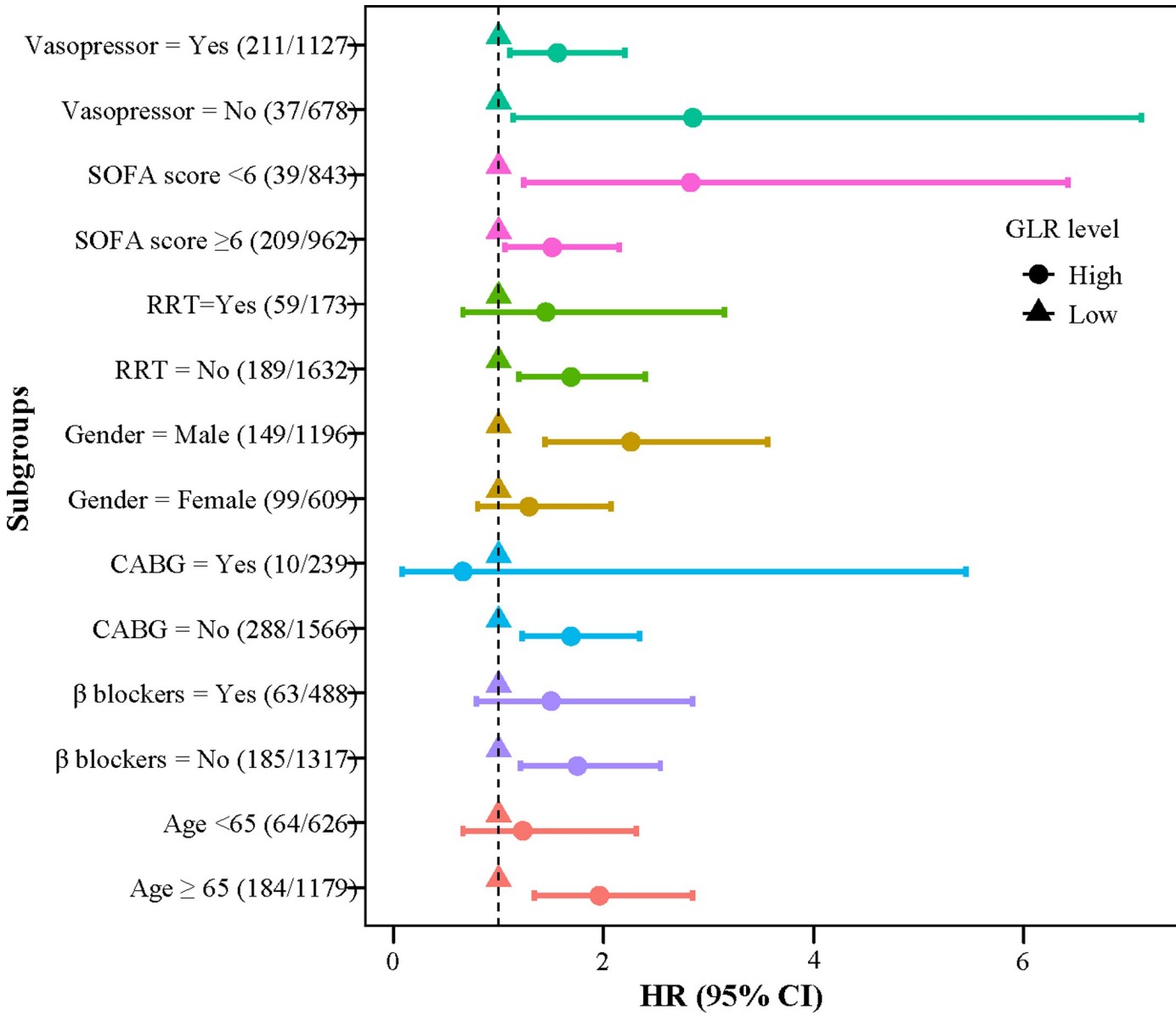

**Fig 4. Association between GLR and in-hospital mortality in subgroups of age, gender, vasopressor use, SOFA scores, RRT, CABG, and β blockers.**

conventional AMI treatment. However, the precise mechanisms of the effect of treatment on the association between GLR and in-hospital mortality in patients with AMI are unclear.

As far as we know, this study is the first to investigate the association between GLR level after ICU admission and the in-hospital mortality risk in patients with AMI as well as the short-term predictive value of GLR on AMI prognosis. GLR has the advantages of being fast, cheap and easy to obtain, which has potential value for rapid identification of high-risk patients in clinical. Nevertheless, there are some limitations in our study. As a single-centre retrospective study, this study inevitably has a particular selection bias. Covariates such as myocardial infarction size which may affect AMI prognosis were not take into consideration because it was unavailable in the MIMIC-IV database. In addition, the sample size of mortality

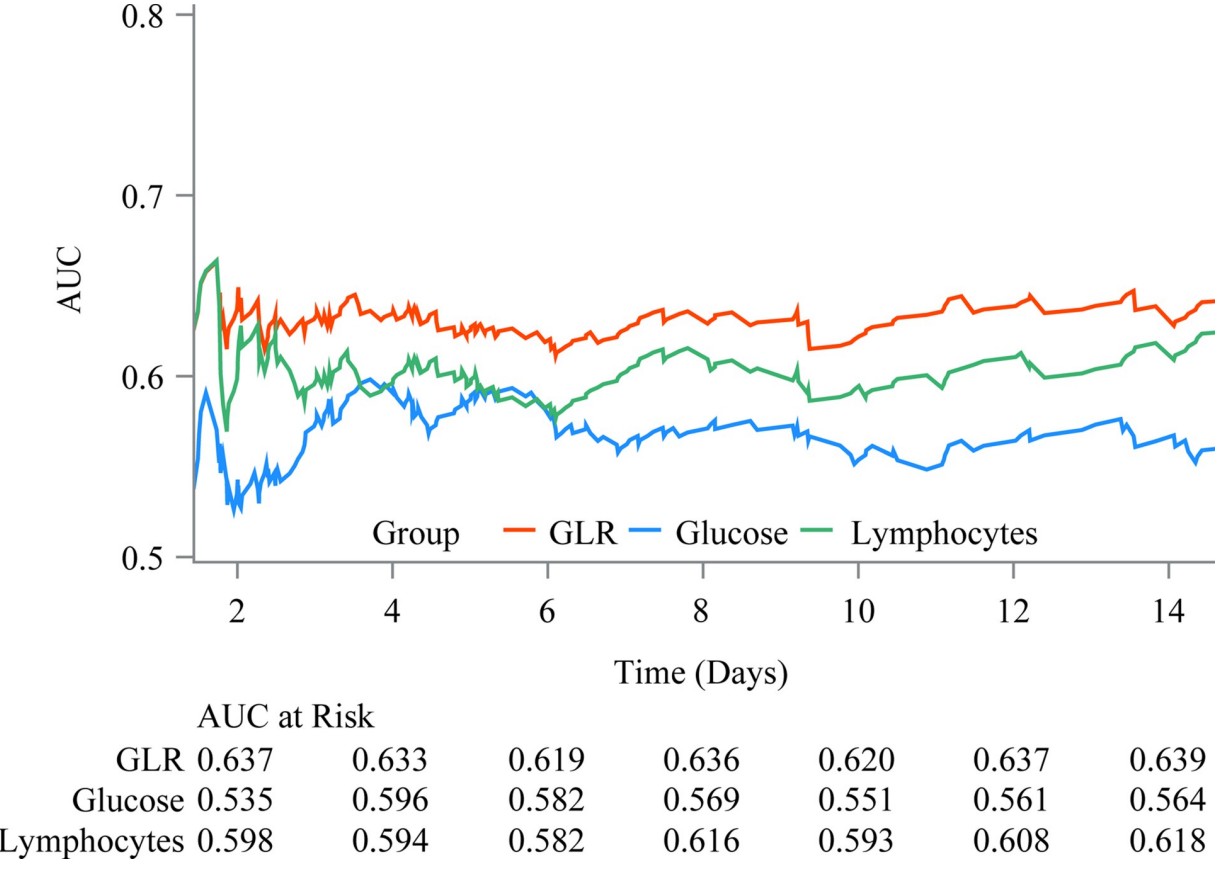

| AUC at Risk | | | | | | | |
| --- | --- | --- | --- | --- | --- | --- |
| GLR | 0.637 | 0.633 | 0.619 | 0.636 | 0.620 | 0.637 | 0.639 |
| Glucose | 0.535 | 0.596 | 0.582 | 0.569 | 0.551 | 0.561 | 0.564 |
| Lymphocytes | 0.598 | 0.594 | 0.582 | 0.616 | 0.593 | 0.608 | 0.618 |

**Fig 5. The predictive value of GLR, glucose, and lymphocytes on in-hospital mortality.**

group in this study is relatively small, so that further prospective cohort studies with large samples are needed to clarify the causal association between GLR and AMI prognosis.

## Conclusion

GLR may be a potential predictor for AMI prognosis, which may help identify and manage the high-risk populations in clinical. However, the causal association between GLR level and mortality risk in AMI is still needed further exploration.

## Supporting information

**S1 File.**
(DOCX)

## Author Contributions

**Conceptualization:** Jing Liu.

**Data curation:** Jing Liu, Xiaogang Hu.

**Formal analysis:** Jing Liu, Xiaogang Hu.

**Investigation:** Jing Liu, Xiaogang Hu.

**Methodology:** Jing Liu, Xiaogang Hu.

**Project administration:** Jing Liu.

**Writing – original draft:** Jing Liu.

**Writing – review & editing:** Jing Liu.

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
