## [Decision Letter · Decision Letter 0]

23 Oct 2023

PONE-D-23-26470Association between glucose-to-lymphocyte ratio and in-hospital mortality in acute myocardial infarction patientsPLOS ONE

Dear Dr. Liu,

Thank you for submitting your manuscript to PLOS ONE. After careful consideration, we feel that it has merit but does not fully meet PLOS ONE’s publication criteria as it currently stands. Therefore, we invite you to submit a revised version of the manuscript that addresses the points raised during the review process.

Please submit your revised manuscript within Dec 07 2023 11:59PM. If you will need more time than this to complete your revisions, please reply to this message or contact the journal office at plosone@plos.org. Please include the following items when submitting your revised manuscript:A rebuttal letter that responds to each point raised by the academic editor and reviewer(s). You should upload this letter as a separate file labeled 'Response to Reviewers'.A marked-up copy of your manuscript that highlights changes made to the original version. You should upload this as a separate file labeled 'Revised Manuscript with Track Changes'.An unmarked version of your revised paper without tracked changes. You should upload this as a separate file labeled 'Manuscript'.If applicable, we recommend that you deposit your laboratory protocols in protocols.io to enhance the reproducibility of your results. Protocols.io assigns your protocol its own identifier (DOI) so that it can be cited independently in the future. For instructions see: https://journals.plos.org/plosone/s/submission-guidelines#loc-laboratory-protocols. Additionally, PLOS ONE offers an option for publishing peer-reviewed Lab Protocol articles, which describe protocols hosted on protocols.io. Read more information on sharing protocols at https://plos.org/protocols?utm_medium=editorial-email&utm_source=authorletters&utm_campaign=protocols.

We look forward to receiving your revised manuscript.

Kind regards,

Nour Amin Elsahoryi, pHD

Academic Editor

PLOS ONE

Journal Requirements:

Reviewers' comments:

**Comments to the Author**

1. Is the manuscript technically sound, and do the data support the conclusions?

Reviewer #1: Yes

Reviewer #2: Yes

2. Has the statistical analysis been performed appropriately and rigorously? 

Reviewer #1: Yes

Reviewer #2: Yes

3. Have the authors made all data underlying the findings in their manuscript fully available?

Reviewer #1: Yes

Reviewer #2: Yes

4. Is the manuscript presented in an intelligible fashion and written in standard English?

Reviewer #1: No

Reviewer #2: Yes

5. Review Comments to the Author

Reviewer #1: The manuscript has potential but it is flawed with a lot of typographical error and needs a lot of editing.

Line 8: Check the spelling of the name of the university

All other comments have been made in the attached document

Reviewer #2: The paper is well-prepared, displaying a solid grounding in its subject matter. The authors have engaged thoroughly with existing literature, providing a clear context for their study. This offers readers a comprehensible background and helps to position the research within the broader academic conversation. The methodology section of the paper is clear and straightforward, showing a systematic approach to the research questions. It appears that careful planning has been applied to ensure that the research process is both robust and reproducible.

The statistical analysis presented in the study is transparent and accessible. The findings seem to be the result of a meticulous analysis process, making them both reliable and valuable for the academic community. This adds a layer of credibility to the paper and enhances its contribution to the field. The conclusion drawn by the authors is engaging. It manages to capture the essence of the research findings and presents them in a way that is both interesting and informative. It provides a solid ending to the paper, leaving the reader with a clear understanding of the significance of the research.

In considering the overall presentation and content of the paper, it appears to be a well-rounded piece of academic work. The simplicity of language and clarity of expression make the paper accessible to a broad audience, ensuring that it can be easily understood and appreciated by both experts in the field and those with a more general interest. Given these considerations, it seems that the paper is a worthy candidate for publication, as it can make a meaningful contribution to ongoing scholarly discussions and investigations within the field.

6. PLOS authors have the option to publish the peer review history of their article (what does this mean?). If published, this will include your full peer review and any attached files.

Reviewer #1: No

Reviewer #2: No

---

## [Author Response · Author response to Decision Letter 0]

31 Oct 2023

Dear Editor and Reviewers,

Thank you very much for your time to review and valuable comments on our manuscript. We have carefully studied all the comments and suggestions, and then revised our paper accordingly. The point-by-point responses to the comments are as follow. We hope our responses adequately address the comments and the revised manuscript can achieve your standards. Thank you for your consideration.

Responses to Reviewer #1’s comments

The manuscript has potential but it is flawed with a lot of typographical error and needs a lot of editing.

Response: Thank you for your approval and valuable comments. We have checked and revised the typographical error in the manuscript, and invited a native English speaker to help us revise the article.

Line 8: Check the spelling of the name of the university

Response: We have revised the name of the university.

All other comments have been made in the attached document

Response: We have accepted all the amendments and revised other mistakes to make the manuscript clearer.

Responses to Reviewer #2’s comments

The paper is well-prepared, displaying a solid grounding in its subject matter. The authors have engaged thoroughly with existing literature, providing a clear context for their study. This offers readers a comprehensible background and helps to position the research within the broader academic conversation. The methodology section of the paper is clear and straightforward, showing a systematic approach to the research questions. It appears that careful planning has been applied to ensure that the research process is both robust and reproducible.

The statistical analysis presented in the study is transparent and accessible. The findings seem to be the result of a meticulous analysis process, making them both reliable and valuable for the academic community. This adds a layer of credibility to the paper and enhances its contribution to the field. The conclusion drawn by the authors is engaging. It manages to capture the essence of the research findings and presents them in a way that is both interesting and informative. It provides a solid ending to the paper, leaving the reader with a clear understanding of the significance of the research.

In considering the overall presentation and content of the paper, it appears to be a well-rounded piece of academic work. The simplicity of language and clarity of expression make the paper accessible to a broad audience, ensuring that it can be easily understood and appreciated by both experts in the field and those with a more general interest. Given these considerations, it seems that the paper is a worthy candidate for publication, as it can make a meaningful contribution to ongoing scholarly discussions and investigations within the field.

Response: Thank you very much for your approval and valuable comments. We have revised the grammatical mistake in the manuscript to make it easier to read and understand.

---

## [Editor Report · Decision Letter 1]

27 Nov 2023

Association between glucose-to-lymphocyte ratio and in-hospital mortality in acute myocardial infarction patients

PONE-D-23-26470R1

Dear Dr. Liu,

We’re pleased to inform you that your manuscript has been judged scientifically suitable for publication and will be formally accepted for publication once it meets all outstanding technical requirements.

Kind regards,

Nour Amin Elsahoryi, pHD

Academic Editor

PLOS ONE
---

## [Editor Report · Acceptance letter]

30 Nov 2023

PONE-D-23-26470R1 

Association between glucose-to-lymphocyte ratio and in-hospital mortality in acute myocardial infarction patients 

Dear Dr. Liu:

I'm pleased to inform you that your manuscript has been deemed suitable for publication in PLOS ONE. Congratulations! Your manuscript is now with our production department. 

Kind regards, 

on behalf of

Dr. Nour Amin Elsahoryi 

Academic Editor

PLOS ONE